# Experimental Analysis and Characterization of High-Purity Aluminum Nanoparticles (Al-NPs) by Electromagnetic Levitation Gas Condensation (ELGC)

**DOI:** 10.3390/nano10102084

**Published:** 2020-10-21

**Authors:** Rana Sabouni Tabari, Mohammad Halali, Akbar A. Javadi, Mohammad Hassan Khanjanpour

**Affiliations:** 1Department of Engineering, University of Exeter, Exeter EX4 4QF, UK; RS683@exeter.ac.uk (R.S.T.); MK592@exeter.ac.uk (M.H.K.); 2Department of Material Science and Engineering, Sharif University of Technology, Tehran 11365-11155, Iran; Halali@sharif.edu

**Keywords:** high-purity Al-NPs, electromagnetic levitation gas condensation, X-ray diffraction, atomic absorption spectroscopy

## Abstract

The production of high-purity aluminum nanoparticles (Al-NPs) is challenging due to the highly reactive nature of Al metals. Electromagnetic levitation gas condensation (ELGC) is a promising method to produce high-purity metallic particles as it avoids the interaction between molten metal and refractory-lined, which guarantees the removal of impurities such as oxygen (O). In this research, high-purity Al-NPs were successfully fabricated via ELGC process and fully characterized. The effects of power input and gas flow rate on particle size and distribution were analyzed using field emission scanning electron microscopy (FESEM), energy dispersive spectroscopy (EDS), and dynamic light scattering (DLS). The results showed that the Al-NPs have spherical morphologies with an average diameter of 17 nm and size distribution of NPs is narrow under helium (He) flow rate of 15 L/min at a constant temperature of 1683 ± 10 K. The purity of the NPs was confirmed by utilizing X-ray diffraction (XRD), atomic absorption spectroscopy (AAS), and X-ray fluorescence (XRF). Finally, metal purity of 99.976% and 99.97% was measured by AAS and XRF analyses, respectively. Moreover, it was found that increasing gas flow rate and sample temperature results in a decrease in the particle size. The particle sizes for the Al-NPs obtained under He atmosphere were smaller than those obtained under Ar atmosphere.

## 1. Introduction

A size-controlled sample of aluminum nanoparticles (Al-NPs) can be used in nanocomposites, transparent conductive coatings, drug delivery, absorbents, heat-transfer fluids, wear-resistant additives, chemical analysis, material surface coatings, biomedical applications, and decorative and reflective materials applications [1,2,3,4]. Producing high-purity Al-NPs not only assists the abovementioned applications but also can be applied by inexpensive equipment. As an active metal, synthesis of high-purity Al-NPs should overcome the oxidation layer of Al particles in the range of 1.7–6 nm [5]. Generally, fabrication of Al-NPs can be carried out in either solid phase, liquid phase, or gas phase. For each of these states, different techniques have been utilized in previous studies. For instance, in the solid phase, mechanical ball milling and mechanochemical techniques; in the liquid phase, solution reduction; and in the gas phase, gas evaporation, exploding wire, and laser ablation processes have been used [6]. Al-NPs are considered hydrogen storage materials [7] and the possibility of complete oxidation of NPs increases with increasing the specific surface area of NPs. Application of this material as additives, propellants, fuels, and explosives is affected by particle size and coating process of Al-NPs. Although reducing the size of NPs is preferable, considering energy release, there are still problems with the oxide layer. A number of publications describe the synthesis processes, and protective coatings of Al, but previous results have not demonstrated high-purity Al content and stability of the particle sizes in a size-controlled distribution of Al-NPs [8,9,10,11]. Li et al. [12] synthesized Al-NPs using the evaporation process for propulsion applications; however, the produced NPs were not spherical in shape, which is an essential stability criterion. Ferrara et al. [13] used microwave plasma processes to produce Al-NPs. The average diameter of the generated NPs by this process was 50 nm, however, the size distribution of NPs was not consistent. For instance, NPs with size of 500 nm were presented in their work. Ghanta et al. [14] synthesized Al-NPs using the arc discharge process by applying direct current between Al electrodes in a liquid environment without the use of vacuum. In 2010, Kermanpur et al. [15] applied the ELGC process in a different experimental setup and Al-NPs with size of 95 nm were synthesized using Ar flow rate of 15 L/min. However, the change of gas flow rates to achieve a narrow size distribution was not taken into consideration. In these methods, controlling the size and distribution of particles and low agglomeration is a major challenge; therefore, the production rate of some of these methods is low and very expensive. 

According to the previous literature, the production of high-purity Al particles has not been investigated in detail. In previous methods, liquid and vapor phase fabrication media have been used to produce Al-NPs, but the final products are not purely Al-NPs and oxide formation of Al-NPs is not preventable [11,12]. The main objective of this study is to synthesize high-purity Al-NPs using electromagnetic levitation gas condensation (ELGC) as an inexpensive and swift method. The key advantage of the ELGC process that attracts the attention of scientists is the lack of interaction between molten metal and refractory-lined, which guarantees the removal of one of the major sources of metal pollution by destructive impurities such as O. Accordingly, pure bulk Al as a charge metal and Ar and He gases as condensation media are utilized in the ELGC process. In this process, levitated drops of molten Al are evaporated and finally converted to NPs by cooling inert gas. The appropriate parameters for the synthesis of Al-NPs are determined. Moreover, the metallic sample, which is prepositioned in an electromagnetic field generated by induction coils of appropriate geometry, is melted and levitated stably [8,9,10,11,12,13,14,15,16]. It should be mentioned that the experimental setup, which is used in this study, does not require costly equipment and can be upscaled to industrial applications with lower investment costs and expenses.

## 2. Methodology 

The melted and levitated sample of a metal can be evaporated at specific temperature and pressure [17]. For producing NPs, the bulk Al sample must be melted. For this purpose, the metal bulk is positioned in the magnetic fields of a levitator. The interaction of the metallic charge with the magnetic fields of the levitator builds up the eddy currents. Such eddy currents, in return, react with the magnetic fields, which generate the needed levitation forces. Simultaneously, the eddy currents produce sufficient heat to melt the bulk Al [17,18,19]. The produced vapor, which is achieved from the melted bulk Al, can be condensed upon a cooling medium while the inert gas is passing on molten metal surface into very small size high-purity metallic particles. The size of the synthesized NPs is affected by droplet temperature, gas flow rate, and thermal properties of the carrier and cooling gas medium [20,21,22,23]. This process has several advantages in producing NPs of materials including the containerless melting of the samples, which leads to high-purity products. In addition, the process is rapid and the size of distribution of NPs is the narrowest among other processes. Producing high-purity NPs without using a container, high rate of production and the possibility of consistent synthesis are among the most important advantages of electromagnetic levitation melting process [22]. There are already some reports, which show that the NPs such as silver, nickel, iron, zinc, iron oxide (Fe_2_O_3_), titanium, and zinc oxide (ZnO) can be produced by the ELGC process [24,25,26]. 

## 3. Experimental Setup

The induction of the magnetic field of high frequency coils causes the sample to overcome the force of gravity and levitation. In this process, the coil’s power to levitate the sample is dependent on its geometry. The suitable design for Al levitation sample includes a conical cylinder with four turns at the lifting part along with five turns wound in the opposite direction at the stabilizing part. The schematic of the coil, which is used for levitation and evaporation of Al, is shown in Figure 1. Herein, the distance between the lifting and stabilizing parts is 16 mm. The particles production chamber, which is made of a quartz (silica) tube with a wall thickness of 1 mm, is attached inside a specially shaped (conical) induction coil for levitation. The coil is connected to a radio-frequency power generator (A 15 kW, 450 kHz radio-frequency generator manufactured by Tapka). For cooling and carrier media, He or Ar is used. This noble gas is passed through a silica gel pathway and then through a tube furnace containing a combination of Al and Cu swarf at 550 °C to keep the levels of O and N gases to acceptable limits (near to zero) and to remove the moisture [7]. 

A schematic of the experimental setup in operation is shown in Figure 2. In this Figure, the levitation system, electromagnetic coil, and particles collection chamber are illustrated. The temperature of the sample is applied by changing the power input of the generator. The temperature of the levitated droplet is measured using a Minolta/Land Cyclops 152 infrared pyrometer and calibration of the emissivity coefficient of the measurement medium is carried out. Accordingly, the impact of temperature of droplets on the NPs is measured. The experimental conditions for the production of Al-NPs are listed in Table 1. In the experimental procedures of synthesis of Al-NPs, the Ar or He gas is fed into the reactor with a given flow rate. Briefly, 1 g of an Al bulk sample with purity of 99.999 wt.% is positioned within the appropriate coil using the withdrawal mechanism of the system. Before running the experiment, the pressure and gas flow rate of the reactor are set to the predefined values (Table 1). The sample should be pulled down the coil immediately when the power supply is turned on. In this stage, the sample is levitated and then it starts to heat up, melt, and vaporize. The Al vapor surrounding the molten droplets is condensed by the Ar or He gas stream flowing upward in the power generator. The condensed particles are finally collected in n-hexane bubble container, which is used to prevent particles from getting agglomerated and oxidized. Three different gas flow rates of Ar and He are applied separately at temperatures above the melting point of Al. Temperature adjustments of samples are accomplished by setting the power generator, and the appropriate temperature is measured as 1683 ± 10 K. The experimental results show that He and Ar flow rates more than 15 L/min lead to a temperature drop in the Al sample melting point because of the limitation of the power generator. It should be noted that the current process takes 3 min for Al to be melted and turned into NPs, which is much faster than similar processes [11,14]. The morphology and structure of the synthesized NPs are characterized by scanning electron microscopy (FESEM Philips X230) and XRD (Bruker, D8ADVANCE with Cu K_α_ anode) and size of the particles is measured by imaging tools. The purity of the Al-NPs is characterized by XRF (Ewai, XD-8010 model) and AAS Spectrometer (PerkinElmer, PinAAcle 900H) analyses. The distribution of NPs is defined by DLS analysis using a Malvern Nano ZS (red badge) ZEN 3600.

## 4. Identification and Characterization of High-Purity NPs

### 4.1. Morphology of the Al-NPs

The FESEM micrographs of the Al-NPs synthesized by the ELGC process using Ar and He flow rates of 10, 15, and 20 L/min under atmospheric pressure and constant temperature of 1683 ± 10 K are shown in Figure 3, Figure 4, Figure 5, Figure 6, Figure 7 and Figure 8 at magnification of 140,000 times and work distance of 19 mm. According to these images, the particles are spherical with sizes of 51, 39, and 49 nm for Ar atmosphere with flow rates of 10, 15, and 20 and 28, 17, and 23 nm for He atmosphere with flow rates of 10, 15, and 20, respectively. It is clearly observed that in atmospheric pressure, by increasing Ar or He flow rates from 10 to 15 L/min, the particle size is reduced and by further increase from 15 to 20 L/min the condensed particles are enlarged and coalesced. However, higher Ar flow rates can lead to turbulent flow, which may cause particle collisions and coalescence in the reactor and trigger larger particle sizes at flow rates beyond 25 L/min. From the obtained images, it is seen that increasing gas flow rate from 15 to 20 L/min at a constant temperature of 1683 ± 10 K, for Ar and He atmospheres, increases particles sizes. Similar results were reported by Wegner et al. [27] who stated that as the number of available atoms in vapor phase increases, the chance of collision between atoms also increases. Figure 6 and Figure 7 show FESEM micrographs of Al particles using He flow rates of 10 and 15 L/min, respectively. It can be seen that using He atmosphere as the cooling medium leads to smaller particle sizes in comparison to Ar atmosphere in which the particle size for flow rate of 10 L/min is 28 nm and for flow rate of 15 L/min is 16.91 nm. Increasing temperature leads to higher degrees of super saturation, and accordingly, the chance of collision among atoms increases and the number of available atoms in vapor phase is also increased [12,22]. This process causes a reduction in critical nucleation radius and particle size. Therefore, to produce spherical Al-NPs of Al with small size and narrow size distribution, the flow rate of He or Ar should be about 15 L/min. It should be noted that the image tools in this study measure the average size of the particles.

The size of the NPs is related to the density and thermal conductivity of the operating gas [24]. The density of the gas with the rate of 10–15 L/min is lower than the rate of 15–20 L/min. Therefore, the cooling rate of the system with flow rate of 10–15 L/min is higher than 15–20 L/min. Accordingly, nanoparticles can move easier to the beaker bubble collector including n-Hexane solution without severe interaction between the atoms. The less interaction between atoms leads to smaller size NPs. On the other hand, at flow rates above 15 L/min, the vapor atoms collide with each other, which causes supersaturation. It results in larger particle sizes and increases the number of available atoms in the n-Hexane beaker bubble collector.

### 4.2. EDS Analysis 

The EDS diagrams of the synthesized Al-NPs under He atmosphere at a constant temperature of 1683 ± 10 K are shown in Figure 9. According to EDS, AAS, and XRF analyses results, the ELGC process can be considered as an appropriate process for producing high-purity Al-NPs. This process has the advantage of being contactless (i.e., no contact between molten drops and container) compared to the other processes of producing Al-NPs. The weight percentage of the Al as the consistent element is shown in Figure 9. The present peak in the spectrum confirms that the sample does not accumulate any other elements except Al. The presence of O in Al-NPs is not detected, which confirms the high-purity level of Al-NPs produced by this method.

### 4.3. Phase and Purity of the Al-NPs

The XRD pattern of the synthesized Al-NPs under He atmosphere at a constant temperature of 1683 ± 10 K and gas flow rate of 15 L/min by this process is shown in Figure 10. The pattern clearly shows the crystalline Al particles. The peaks correspond to pure Al (JCPCS card #00-004-0787). The XRD measurements are carried out on powder sample of Al-NPs using an X-ray particles Diffractometer Bruker, D8ADVANCE with Cu K_α_ radiation (λ = 0.154056 nm). The analysis is accomplished over the 2θ range from 5° to 120°^,^ at a step increment of 0.02° and scan speed of 3° per step for 4 times. According to the obtained results in Figure 10, the purity of the Al-NPs is characterized and it shows a high level of purity of Al-NPs, and there are no peaks relevant to any kind of crystalline Al oxides. However, for 2θ less than 38°, as the sample holder is made of quartz, a small amount of this material is analyzed, which can be ignored [24]. This is in contrast to other similar works that characterized Al_x_O_y_ phases in their XRD analysis [12,14,28,29,30]. The chemical composition and structure of Al-NPs is determined by the corresponding XRD pattern. This pattern includes eight sharp peaks for Al-NPs as (2θ = 38.56°, 44.79°, 65.12°, 78.20°, 82.41°, 99.06°, 111.98°, and 116.52°) relevant to Miller indices of (110), (200), (220), (311), (222), (400), (331), and (420), respectively (Table 2). The particles are face-centered cubic (FCC) pure Al. There is no peak for the other elements or the Al oxides. After assessing the XRD of the obtained NPs (Figure 10), a complete match with the reference pattern (JCPCS card #00-004-0787) is observed, which confirms the formation of these NPs. However, the current XRD pattern for our sample is wider in peaks as the particles are smaller and in nanoscale, based on Williamson–Hall equation. 

AAS analysis is carried out for Al-NPs and the results show metal purity of 99.976% wt.%, and XRF analysis shows Al purity of +99.97% wt.% (Figure 11 and Table 3). Other elements than Al, including Mg, Mn, Fe, Si, and Cu are detected by AAS, which are around 0.025 wt.%. The Al-NPs are washed in dichloromethane for 12 min and organic contamination is removed by suction filter from the surface (Table 3). In addition, the XRF spectrometer is applied to analyze the sample. The X-ray tube works as a stimulation source, and the exposure of the plasma shows the major element concentration as Al with intensity of +99.97%. Herein, the region of 2.5–10 KeV is negligible for the diffraction lines, which are not fluorescence from the Al-NPs for the XRF result. Accordingly, for the sample containing less than 1% of Mg, Mn, and Fe the diffraction lines are not illustrated. It should be mentioned that the determination of the main components of Al takes 200 s in total. Some of the final levels of purity of previously produced Al-NPs are listed in Table 4. It can be seen that the purity of the current study (99.976%) is higher than the mentioned works. 

### 4.4. Particle Size Distribution of Produced Al-NPs

For DLS analysis, the sample of NPs is dispersed in deionized water. The dispersibility and particle size distribution of the synthesized Al-NPs are defined using DLS analysis (Figure 12). According to this Figure, an appropriate dispersed state of production is in the range of 9–28 nm for He atmosphere and from 18 to 57 nm for Ar atmosphere. The mean particle size diameter is about 18.18 nm prepared under He atmosphere at a temperature of 1683 ± 10 K and 38 nm under Ar atmosphere. The size distribution of the particles is unchanged even after 90 days of repeating DLS analysis and the dispersibility of NPs is unified in deionized water. The thermal conductivity of He is 8 times more than Ar [34]. He cools faster than Ar and needs less time for coalescence of particles due to its high thermal conductivity. On the other hand, as Ar atmosphere is heavier than He, the chance of agglomeration and coalescence of Ar is higher than He [35]. The obtained results for the particle sizes of the produced NPs under He and Ar media are in good agreement with similar research works [22,24], in producing silver and titanium NPs by the ELGC process. Moreover, DLS analysis confirms the FESEM outcomes for narrow particle size distribution under 15 L/min flow rate of Ar and He as cooling media. It can be seen that the mean sizes of Al-NPs synthesized by this process are smaller under He atmosphere and are about 17 and 38 nm for He and Ar atmospheres, respectively, and the standard deviation values for these diameters are around 11 nm for He and 29 nm Ar atmospheres (Table 5). In terms of the direct relationship between the atomic mass of carrier gas and the average particle size, a heavier carrier gas (Ar) can obtain higher collisions between gas atoms and Al atoms. However, since thermal conductivity of He is higher than Ar, the cooling power of He is more than Ar atmosphere, which leads to smaller particle sizes without any further chance for particles to coalesce and grow.

The size and distribution of particles and flow rates of Ar and He are summarized in Figure 13. It can be seen that the size of particles for Ar atmosphere for flow rate less than 15 L/min in constant temperature is larger than He. For both gas atmospheres in 15 L/min flow rate, there is a noticeable decrease in size of particles (a drop of 23.53% for Ar and 39.30% for He atmospheres) and distribution of particles is homogenous. However, an increase in flow rate to 20 L/min causes growing of particles, and the particle size is grown for both atmospheres of Ar and He simultaneously.

## 5. Conclusions

In the present study, high-purity Al-NPs were synthesized using an ELGC process. The effects of input power and gas flow rate on particle sizes and their size distribution were investigated using FESEM, EDS, and DLS analyses. It was found that the appropriate flow rate was 15 L/min for both Ar and He atmospheres at a constant temperature of 1683 ± 10 K. The FESEM images and EDS results illustrated that the increase in gas flow rate from 10 to 15 L/min at a constant temperature results in a decrease in the size of the produced Al-NPs. However, gas flow rates higher than 15 L/min may lead to an increase in the size of NPs, and collision could occur consequently. The XRD pattern clearly showed the formation of crystalline Al particles, which is in a good agreement with the published literature. In addition, the current XRD pattern of sample is wider due to the formation of nanoparticles and no peaks relevant to Al oxides element were observed. The purity of samples was analyzed by both AAS and XRF. Al was the major element with purity of 99.976% from AAS and 99.97% from XRF. The produced Al-NPs sample exhibits a medium particle size of 17.4 nm under He atmosphere at a constant temperature of 1683 ± 10 K, which is close to the particle size obtained from FE-SEM images (17 nm). In addition, helium is more effective than argon in the production of small size Al-NPs with narrower size distribution.

## Figures and Tables

**Figure 1 nanomaterials-10-02084-f001:**
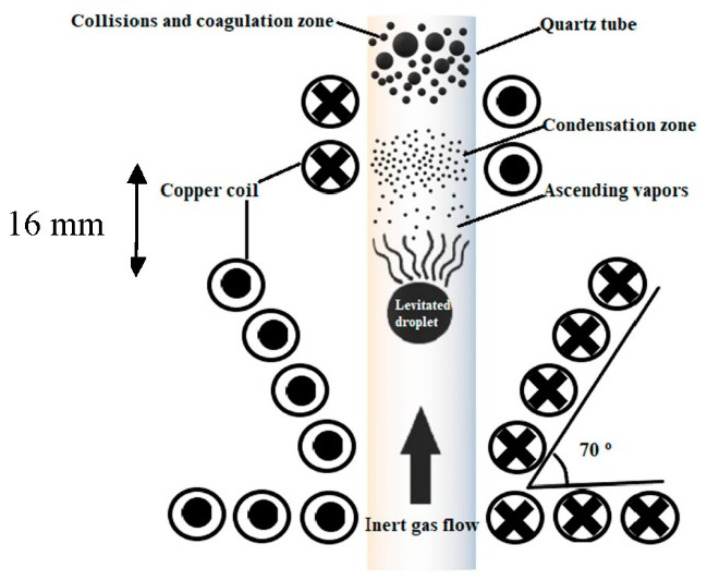
Schematic of coil for electromagnetic levitation gas condensation (ELGC) setup.

**Figure 2 nanomaterials-10-02084-f002:**
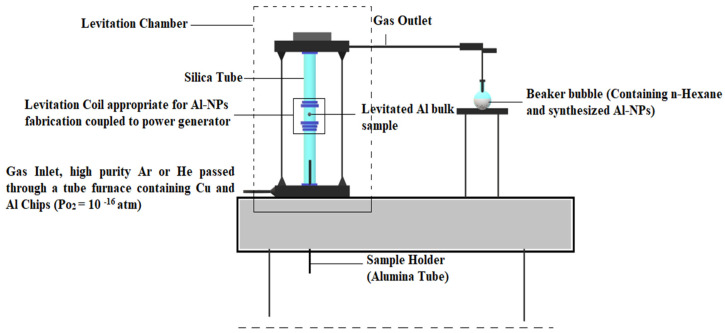
Schematic diagram of ELGC setup. The conical coil is coupled to a power generator and aluminum (Al) bulk sample is melted, levitated, and evaporated in the electromagnetic field [18].

**Figure 3 nanomaterials-10-02084-f003:**
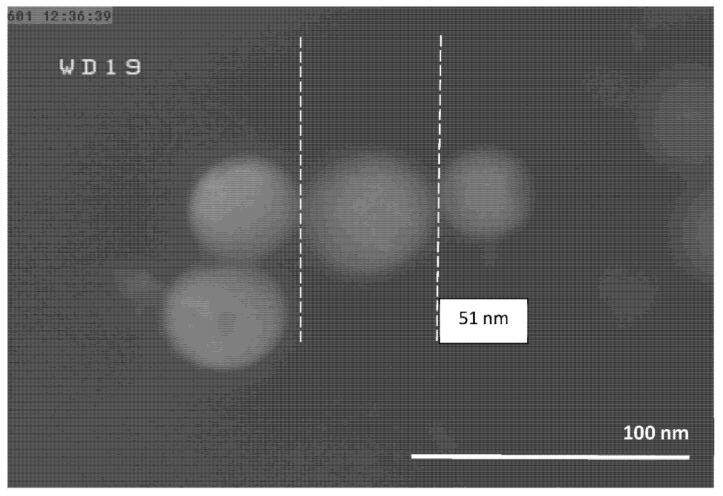
FESEM micrograph of Al particles using Ar flow rate of 10 L/min for magnification 140 kx.

**Figure 4 nanomaterials-10-02084-f004:**
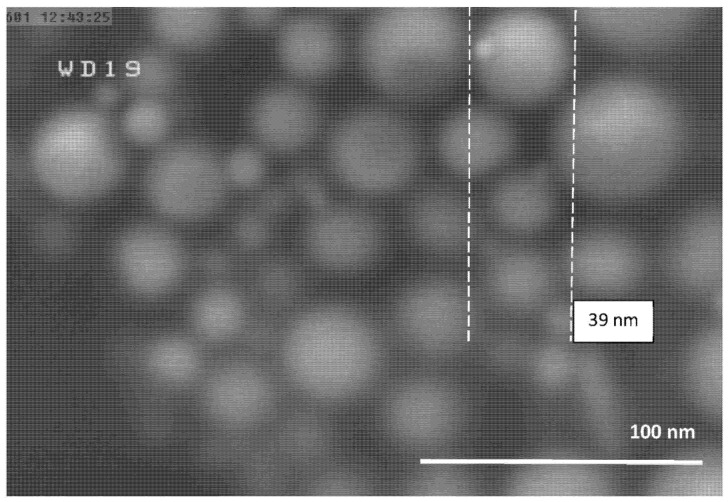
FESEM micrograph of Al particles using Ar flow rate of 15 L/min for magnification 140 kx.

**Figure 5 nanomaterials-10-02084-f005:**
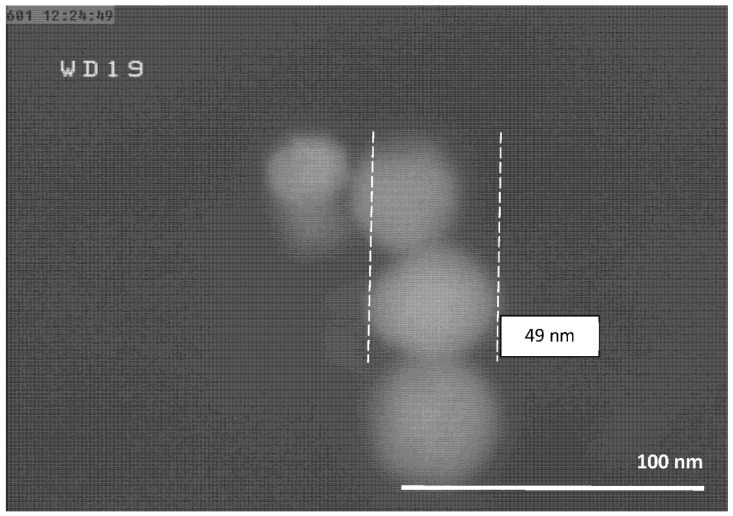
FESEM micrograph of Al particles using Ar flow rate of 20 L/min for magnification 140 kx.

**Figure 6 nanomaterials-10-02084-f006:**
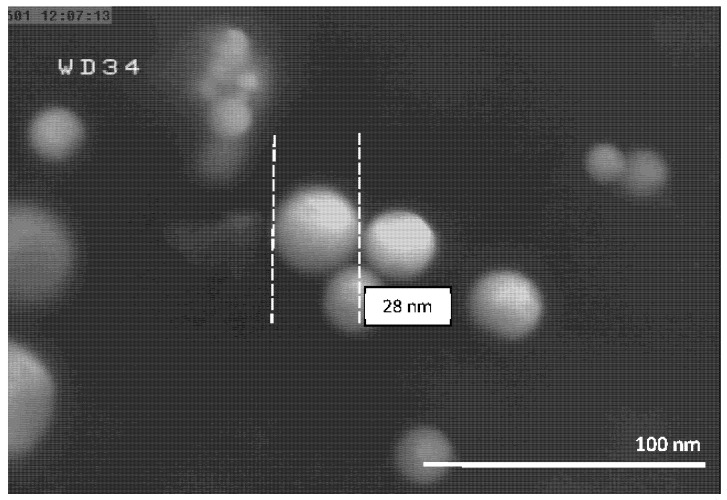
FESEM micrograph of Al particles using He flow rate 10 L/min for magnification 140 kx.

**Figure 7 nanomaterials-10-02084-f007:**
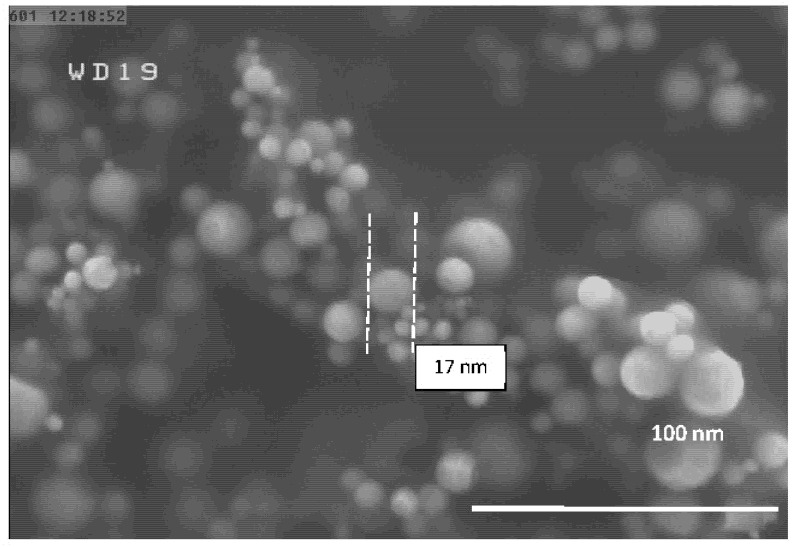
FESEM micrograph of Al particles using He flow rate of 15 L/min for magnification 140 kx.

**Figure 8 nanomaterials-10-02084-f008:**
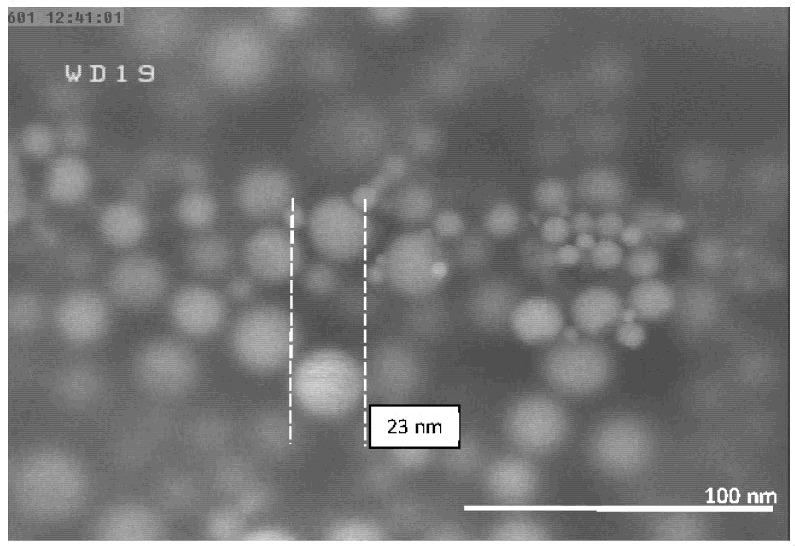
FESEM micrograph of Al particles using He flow rate of 20 L/min for magnification 140 kx.

**Figure 9 nanomaterials-10-02084-f009:**
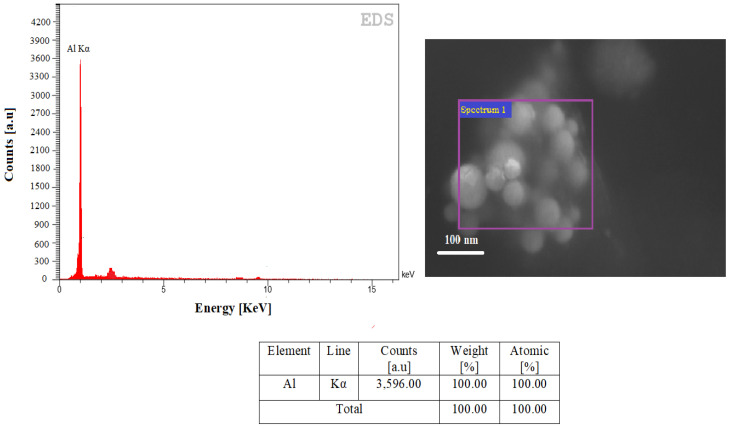
The EDS elemental energy spectra illustration of clear Al peak.

**Figure 10 nanomaterials-10-02084-f010:**
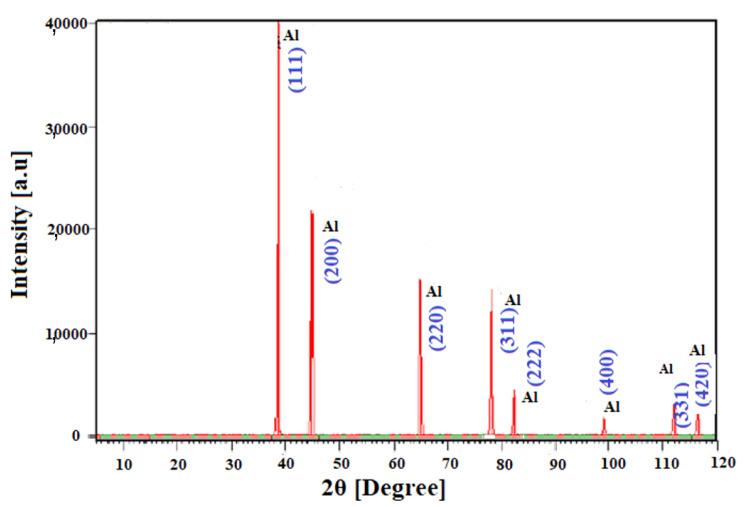
The XRD pattern of aluminum nanoparticles (Al-NPs) produced by ELGC method.

**Figure 11 nanomaterials-10-02084-f011:**
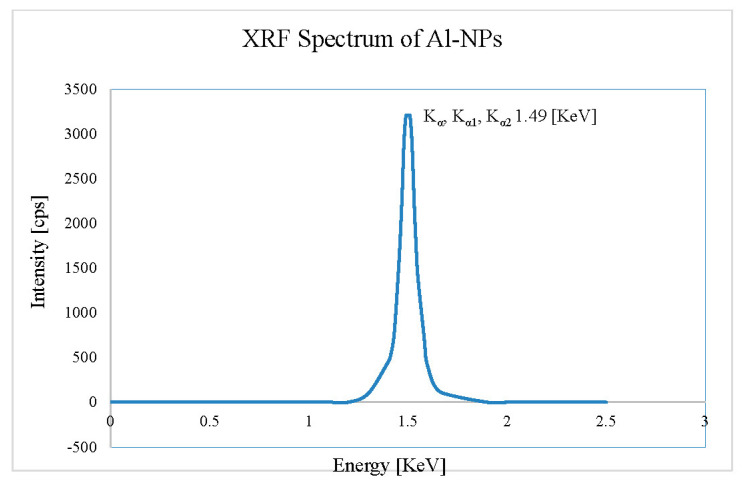
XRF analysis graph of Al-NPs by ELGC method.

**Figure 12 nanomaterials-10-02084-f012:**
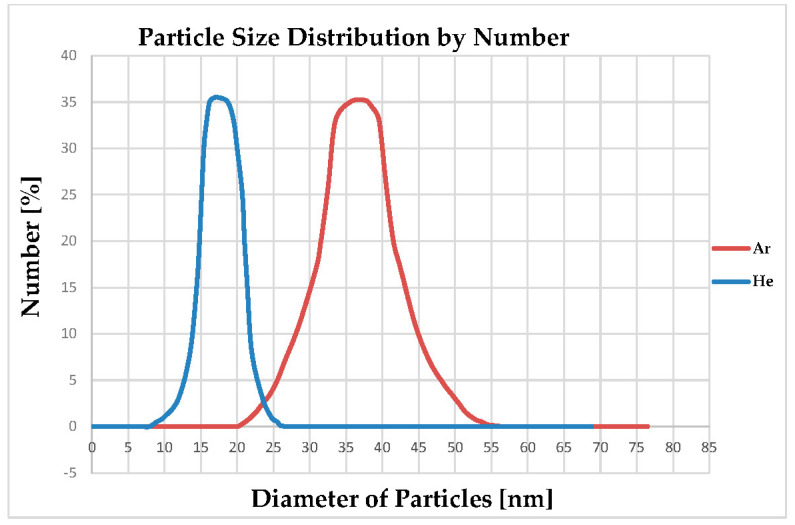
Dynamic light scattering (DLS) diagram of NPs size distribution under Ar and He atmospheres for 15 L/min at a constant temperature of 1683 ± 10 K.

**Figure 13 nanomaterials-10-02084-f013:**
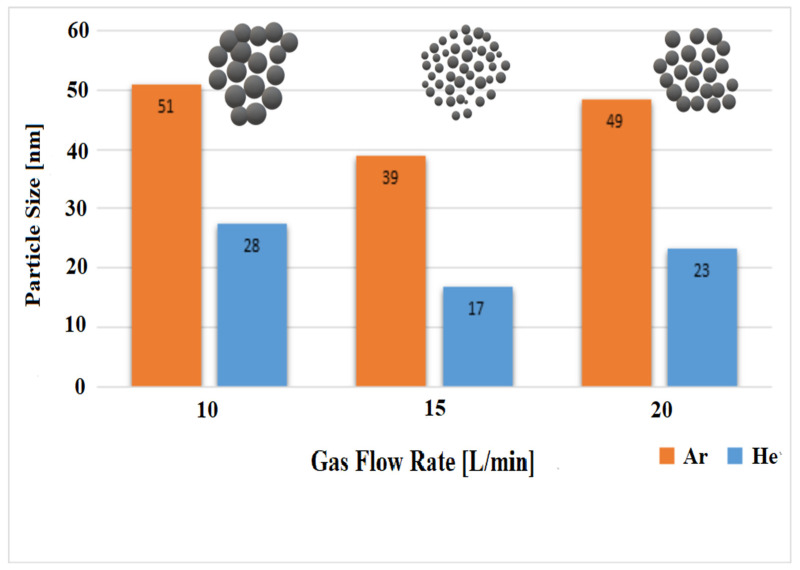
A graphic comparison between two gases as cooling atmospheres on the particle sizes of produced Al-NPs.

**Table 1 nanomaterials-10-02084-t001:** Experimental conditions for synthesis of aluminum (Al).

Experiment Number	Ar Flow Rate (L/min)	Temperature of Al Droplet (K)	Particle Size (nm)
1	10	1683 ± 10	51
2	15	1683 ± 10	39
3	20	1683 ± 10	49
**Experiment Number**	**He Flow Rate (L/min)**	**Temperature of Al Droplet (K)**	**Particle Size (nm)**
4	10	1683 ± 10	28
5	15	1683 ± 10	17
6	20	1683 ± 10	23

**Table 2 nanomaterials-10-02084-t002:** Quantitative XRD analysis of pure aluminum nanoparticles (Al-NPs).

2θ (Degree)	Intensity (a.u)	d-Spacing (A°)	Plane		
38.5624	40,404.19	2.33472	1	1	1
44.7953	21,916.62	2.02328	2	0	0
65.1216	16,239.63	2.43245	2	2	0
78.2060	14,216.02	1.22232	3	1	1
82.4127	4451.56	1.17026	2	2	2
99.0613	1887.77	1.0346	4	0	0
111.9845	3151.97	0.93000	3	3	1
116.5297	2654.67	0.90571	4	2	0

**Table 3 nanomaterials-10-02084-t003:** Atomic absorption spectroscopy (AAS) results of element content.

Component	Symbol	Concentration Range Mass (%)
Aluminum	Al	99.976
Magnesium	Mg	0.009
Manganese	Mn	0.009
Iron	Fe	0.003
Silicon	Si	0.001
Copper	Cu	0.001

**Table 4 nanomaterials-10-02084-t004:** Maximum levels of purity of previous studies.

Researcher(s)	Content of Al (wt.%)
Gromov et al. [31]	90
Gazanfari et al. [11]	92.5
Lin et al. [32]	99.0
Mahendiran et al. [33]	99.0

**Table 5 nanomaterials-10-02084-t005:** Dynamic light scattering (DLS) measurement of Al-NPs suspended in deionized water.

Ar Atmosphere	Diameter (nm)	Intensity (%)	Standard Deviation (nm)	PDI
Peak	38	94.2	29	0.567
**He Atmosphere**	**Diameter (nm)**	**Intensity (%)**	**Standard Deviation (nm)**	**PDI**
Peak	17	93.7	11	0.442

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
