# Peer review of "Experimental Analysis and Characterization of High-Purity Aluminum Nanoparticles (Al-NPs) by Electromagnetic Levitation Gas Condensation (ELGC)"

_nanomaterials, 2020, doi:10.3390/nano10102084_

Round 1

Reviewer 1 Report

Comments to Author:

Ms. Ref. No.: nanomaterials-966883

Title: Experimental analysis and characterization of high-purity aluminum nanoparticles (Al-NPs) by Electromagnetic Levitation Gas Condensation (ELGC)

Overview and general recommendation:

  1. The current study is on a topic of relevance and general interest to the readers of the journal with a relative impact on scientific research.
  2. In general, a clear thread should be woven through the text linking the introduction, methods, results, and conclusions. After carefully checking the article, I would suggest the following improvements to the article.
  • First of all, please review the article for the English language. Some phrases/sentences/ are not clear, some words are not properly used.
  • The authors clearly prove that they do not know the IUPAC requirements regarding the correct writing of the unit of measurements. It is not possible to use in the scientific articles as UM lit/min! Really, was a huge surprise for me! In the whole article is the same mistake of UM! The correct UM is L for Liter, so the authors must use L/min! Another important mistake is UM of temperature! Never, but never do not use degree when it is Kelvin. The symbol for Kelvin is K not 0 This is another important requirement of IUPAC and IUPAF for UM. These mistakes, from my point of view, lead to the rejection of the article!
  • Watch the use of present tense vs past tense. There is not a clear tense throughout the manuscript. Typically, the present tense is used when discussing the topic research, but there are cases where the past can be appropriate.
  • Figure 2 must be deleted! Only in the Master of PhD thesis are used these figures!
  • FE-SEM-EDS interpretation has a lot of mistakes as well.
  • All the figures must be replaced with others at a better resolution.
  1. The novelty of the study is not clearly explained. There are many articles with the same content, for different parts of the world. What is the difference between their content and that of the present paper? The aim of this work at the end of the introduction is unclear and must be correlated with the novelty of this study. Please provide a clear objective/hypothesis in the Introduction Section and Abstract as well!
  2. Again, the aim of this work in the abstract and the end of the introduction must be correlated with the novelty of this study. Please highlight the importance of this research both in the Abstract and the Conclusions.
  3. Conclusions section - please improve this section with the novelty of this study! Explain how other readers can use this work in the future!
  4. Typically, the beginning of a sentence with which, when, and so on are not accepted in scientific writing. Please check in the whole article!
  5. From my point of view, the discussions must be improved and rearranged being correlated with the experimental part. To increase the scientific level of this article for publication, I recommend relevant discussions in this regard. This section could be a bit clearer.
  6. The conclusion Section must be reformulated. In the conclusion section, there is no hypothesis that would greatly help focus and clarify the results of this manuscript! 
  7. The abstract must rewrite!
  8. More references must be inserted! A well scientific article must have over 50 references!
  9. In conclusion, this article does not respect the requirements/template of the journal!
  10. Please consider having your paper reviewed by a native English speaker for language quality!
  11. Rather, I recommend this study to be submitted to other journals, not Nanomaterials!

Author Response

Reviewer #1: In general, a clear thread should be woven through the text linking the introduction, methods, results, and conclusions. After carefully checking the article, I would suggest the following improvements to the article.

1) First of all, please review the article for the English language. Some phrases/sentences/ are not clear, some words are not properly used.

Response: The paper was thoroughly revised for quality of English, as suggested.

2) The correct UM is L for Liter, so the authors must use L/min! Another important mistake is UM of temperature! Never, but never do not use degree when it is Kelvin. The symbol for Kelvin is K not 0 This is another important requirement of IUPAC and IUPAF for UM.

Response: All the units were checked and revised as suggested. Especially lit/min was changed to L/min, and oK was changed to K.

3) Watch the use of present tense vs past tense. There is not a clear tense throughout the manuscript. Typically, the present tense is used when discussing the topic research, but there are cases where the past can be appropriate.

Response: The paper was revised for the use of tense and grammar.

4) Figure 2 must be deleted.

Response: Figure 2 was deleted and a new schematic diagram was replaced.

5) All the figures must be replaced with others at a better resolution.

Response: The figures are replaced with others with better resolution.

6) The novelty of the study is not clearly explained. There are many articles with the same content, for different parts of the world. What is the difference between their content and that of the present paper? The aim of this work at the end of the introduction is unclear and must be correlated with the novelty of this study. Please provide a clear objective/hypothesis in the Introduction Section and Abstract as well!

Response: The following paragraph was added (Line 66) to address this comment.

“According to the previous literature, the production of high-purity Al particles has not been investigated in detail. In previous methods, liquid and vapor phase fabrication media have been used to produce Al-NPs, but the final products are not purely Al-NPs and oxide formation of Al-NPs is not preventable [11, 12]. The main objective of this study is to synthesize high-purity Al-NPs using Electromagnetic Levitation Gas Condensation (ELGC) as an inexpensive and swift method. The key advantage of the ELGC process that attracts the attention of scientists is the lack of interaction between molten metal and refractory-lined, that guarantees the removal of one of the major sources of metal pollution by destructive impurities such as oxygen. Accordingly, pure bulk Al as a charge metal, and Ar and He gases as condensation media are utilized in the ELGC process. In this process, levitated drops of molten Al are evaporated and finally converted to NPs by cooling inert gas. The appropriate parameters for the synthesis of Al-NPs are determined. Moreover, the metallic sample, which is pre-positioned in an electromagnetic field generated by induction coils of appropriate geometry, is melted and levitated stably [8-16]. It should be mentioned that the experimental set-up, which is used in this study, does not require costly equipment and can be upscaled to industrial applications with lower investment costs and expenses.“

7) Conclusions section - please improve this section with the novelty of this study! Explain how other readers can use this work in the future!

Response: The Conclusion section was revised as follows:

“In the present study, high-purity Al-NPs were synthesized by using an ELGC process. The effects of input power and gas flow rate on particle sizes and their size distribution were investigated by using FESEM, EDS, and DLS analyses. It was found that the appropriate flow rate was 15 L/min for both Ar and He atmospheres at a constant temperature of 1683±10 K. The FESEM images and EDS results illustrated that the increase in gas flow rate from 10 to 15 L/min at a constant temperature results in a decrease in the size of the produced Al-NPs. However, gas flow rates higher than 15 L/min may lead to an increase in the size of NPs, and collision could occur consequently. The XRD pattern clearly showed the formation of crystalline Al particles, which is in a good agreement with the published literature. In addition, the current XRD pattern of sample is wider due to the formation of nanoparticles and no peaks relevant to Al oxides element were observed. The purity of samples was analyzed by both AAS and XRF. Al was the major element with purity of 99.976% from AAS and 99.97% from XRF. The produced Al-NPs sample exhibits a medium particle size of 17.4 nm under He atmosphere at a constant temperature of 1683±10 K, which is close to the particle size obtained from FE-SEM images (17 nm). In addition, helium is more effective than argon in the production of small size Al-NPs with narrower size distribution.”

8) Typically, the beginning of a sentence with which, when, and so on are not accepted in scientific writing.

Response: The paper was thoroughly revised for quality of English.

9) The abstract must rewrite!

Response: The Abstract was revised as follows:

“The production of high-purity aluminum nanoparticles (Al-NPs) is challenging due to the highly reactive nature of Al metals. Electromagnetic Levitation Gas Condensation (ELGC) is a promising method to produce high-purity metallic particles as it avoids the interaction between molten metal and refractory-lined, which guarantees the removal of impurities such as oxygen. In this research, high-purity Al-NPs were successfully fabricated via ELGC process and fully characterized. The effects of power input and gas flow rate on particle size and distribution were analyzed using field emission scanning electron microscopy (FESEM), energy dispersive spectroscopy (EDS), and dynamic light scattering (DLS). The results showed that the Al-NPs have spherical morphologies with an average diameter of 17 nm and size distribution of NPs is narrow under helium (He) flow rate of 15 L/min at a constant temperature of 1683±10 K. The purity of the NPs was confirmed by utilizing X-ray diffraction (XRD), atomic absorption spectroscopy (AAS), and X-ray fluorescence (XRF). Finally, metal purity of 99.976% and 99.97 % was measured by AAS and XRF analysis respectively. Moreover, it was found that increasing gas flow rate and sample temperature results in a decrease in the particle size. The particle sizes for the Al-NPs obtained under He atmosphere were smaller than those obtained under Ar atmosphere.”

10) More references must be inserted! A well scientific article must have over 50 references!

Response: Some relevant references were reviewed and added to the paper.

11) Please consider having your paper reviewed by a native English speaker for language quality!

Response: The paper was thoroughly revised for quality of English.

Reviewer 2 Report

Dear Authors,

Sincerely,

The referee

Author Response

Reviewer #2: This paper concerns the production method of high-purity aluminum nanoparticles. The paper is well-written for the readers but there are plenty of grammatical mistakes.

1) As a major point, the authors should mention how to observe the nano-particles using FESEM. Where did the authors put the particle?

Response: The fabricated NPs were collected in a beaker bubble including n-Hexane as a liquid medium. As particles were in n-Hexan solution and we wanted to avoid oxidation of the particles in the solution, centrifuge technique was applied to separate NPs from the residual solution and later on cryo-FESEM preparation technique was done on each sample. The sample was moved into the FESEM chamber under vacuum which was located on a cold stage.

2) The photos show very clear background, and the result of EDS analysis shows only the signal of Al without any artifact. How the authors avoided the aggregation of nanoparticles?

Response: To prevent aggregation of NPs, three flow rates were applied at a constant temperature and it was observed that for flow rates more than 20 L/min, NPs started to be agglomerated and grown in size. Therefore, to control the conditions, as explained in the paper, there was no physical interactions between nanoparticles, especially for flow rate of 15 L/min in both atmospheres of Ar and He. In addition, aggregation is possible when the sample is dried. However, in this study, the original state of samples was kept in liquid medium for experimental analysis.

3) L43; “in range of 1.7” was changed to “in the range of 1.7 to”.

L44; “these states different techniques” was changed to “these states, different techniques”.

L52; “evaporation process for” was changed to “the evaporation process for”.

L55; “was not consist” was changed to “was not consistent”.

L91; “containerless melting” was changed to “the containerless melting”.

L101; “high frequency” was changed to “high-frequency”.

L106; “which is made of quartz (silica) tube” was changed to “which is made of a quartz (silica) tube”. L123; “conditions for production of Al-NPs” was changed to “conditions for the production of Al-NPs”.

L140; “Purity of the Al-NPs is” was changed to “The purity of the Al-NPs is”.

L163; “between each atoms” was changed to “between each atom”.

L228; “However the current” was changed to “However, the current”.

L247; “purity of current study” was changed to “purity of the current study”.

Reviewer 3 Report

Herein R.S. Tabari et al proposed and characterized Al NP synthesized through their Electromagnetic Levitation Gas Condensation (ELGC) processes. The authors improved significantly from ELGC process described in 2010, with finer NP size, purity and production rate. Characterized through FESEM, EDS & DLS, their Al NP reached ~17nm with He gas at 15 lit/min. Meanwhile, XRD, AAS & XRF, revealed a purity of 99.97% Al.

Overall, I believe the article is of significant impact and worth publishing in Nanomaterials, subject to adequate clarification/revision noted below:

  • The authors mentioned their fabrication method results in narrow size distribution. I suggest to corroborate size distribution profile of DLS with that of FESEM result? In particular, FESEM results in Fig 7&8 shows quite a wide distribution of NP.
  • From 15 to 20 lit/min Ar/He flow rate, obtained NP size increased, which the authors attributed to the increased collision between atoms. Suggest the authors to explain why size reduction occurred for 10 to 15 lit/min flow rate.
  • Why was the temperature fixed at 1683 K? The authors mentioned increasing temperature can lead to smaller NP - has the authors adjusted this temperature alongside the gas flow rate for finer NP?
  • Please improve the resolution for Fig 11B.
  • Suggest the authors to put polydispersity index (PDI) values alongside DLS distribution in Fig 12.

Author Response

Reviewer #3: Overall, I believe the article is of significant impact and worth publishing in Nanomaterials, subject to adequate clarification/revision noted below:

1) The authors mentioned their fabrication method results in narrow size distribution. I suggest to corroborate size distribution profile of DLS with that of FESEM result? In particular, FESEM results in Fig 7&8 shows quite a wide distribution of NP.

Response: Under the He atmosphere, the density of the fabricated particles will be increased in comparison with the Ar atmosphere. Therefore, the area including the NPs shows a wider range of size distribution, and sizes of NPs are decreased under He atmosphere. As we presented the FESEM images of all samples in one magnification, Figures 7 and 8 demonstrate the difference of these two atmospheres specifically. According to the results, the range of particle size covers sizes from 7 nm to 24 nm for Figure 7 and from 2 nm to 35 nm for Figure 8. However, most of the particle sizes are close to the average size of 17 nm in Figure 7 and 23 nm in Figure 8. It must be noted that both Figures are allocated to He atmosphere, as the mass of distributed NPs in Ar atmosphere for a specific area is less than He atmosphere. The following sentence was added in Line 290 to clarify this comment:

“It can be seen that the mean sizes of Al-NPs synthesized by this process are smaller under He atmosphere and are about 17 nm and 38 nm for He and Ar atmospheres, respectively, and the standard deviation values for these diameters are around 11 nm for He and 29 nm Ar atmospheres.”

2) From 15 to 20 lit/min Ar/He flow rate, obtained NP size increased, which the authors attributed to the increased collision between atoms. Suggest the authors to explain why size reduction occurred for 10 to 15 lit/min flow rate.

Response: The following paragraph was added (Line 209) to address this comment:

“The size of the NPs is related to the density and thermal conductivity of the operating gas [24]. The density of the gas with the rate of 10-15 L/min is lower than the rate of 15-20 L/min. Therefore, the cooling rate of the system with flow rate of 10-15 L/min is higher than 15-20 L/min. Accordingly, nanoparticles can move easier to the beaker bubble collector including n-Hexan solution without severe interaction between the atoms. The less interaction between atoms leads to smaller size NPs. On the other hand, at flow rates above 15 L/min, the vapor atoms collide with each other which causes supersaturation. It results in larger particle sizes and increases the number of available atoms in the n-Hegxan beaker bubble collector.”

3) Why was the temperature fixed at 1683 K? The authors mentioned increasing temperature can lead to smaller NP - has the authors adjusted this temperature alongside the gas flow rate for finer NP?

Response: This temperature was obtained by trial and error. The target temperature was to achieve levitation and melting of the bulk sample as well as evaporation and condensation of particles simultaneously. Yes, this temperature was appropriate for both gases to gain finer NPs. 

4) Please improve the resolution for Fig 11B.

Response: The resolution of Figure 11 was improved as suggested.

5) Suggest the authors to put polydispersity index (PDI) values alongside DLS distribution in Fig 12.

Response: The PDI values were added to Figure 12 as suggested.

Round 2

Reviewer 1 Report

Comments to Author:

Ms. Ref. No.: nanomaterials-966883

Title: Experimental analysis and characterization of high-purity aluminum nanoparticles (Al-NPs) by Electromagnetic Levitation Gas Condensation (ELGC)

Overview and general recommendation:

The Manuscript was improved but still has a lot of mistakes. E.g.,

  • Line 131: In the sentence “Al and cooper…. Oxygen and nitrogen…” the Al must be replaced with aluminum or copper with the chemical symbol Cu, oxygen with O, and so on. Please be careful with these symbols in the whole article.
  • In figures 9 and 10 the UM must be in square brackets [%], [Degree], [a.u], and so on (i.e., in figures and tables as well).
  • The same observation for figure 11. Please improve the quality of this figure. I recommend splitting the table and figure with a strong reference/explanation in the main text!
  • Tables 2 and 3 - UM must be in square brackets.
  • Figure 13 - UM must be in square brackets. Please check in all figures!
  • After figure 13 must be a relevant paragraph or more with a relevant explanation to this figure!
  • I recommend a new spelling for English!

Author Response

Response to Reviewer 1’ comments (nanomaterials-966883)

The Manuscript was improved but still has a lot of mistakes. E.g.,

1) Line 131: In the sentence “Al and cooper…. Oxygen and nitrogen…” the Al must be replaced with aluminum or copper with the chemical symbol Cu, oxygen with O, and so on. Please be careful with these symbols in the whole article.

Response: All mentioned elements were replaced with their related chemical symbols.

2) In figures 9 and 10 the UM must be in square brackets [%], [Degree], [a.u], and so on (i.e., in figures and tables as well).

Response: The UM of Figures 9 and 10, 13 and Tables 2 and 3 were placed in square brackets.

3) The same observation for figure 11. Please improve the quality of this figure. I recommend splitting the table and figure with a strong reference/explanation in the main text!

Response: Figure 11 is replaced with another figure with better resolution. The Figure 10 is also split to one table and one figure.

4) After figure 13 must be a relevant paragraph or more with a relevant explanation to this figure!

Response: The explanation is added in Line 320.
